# Viability testing and transplantation of marginal livers (VITTAL) using normothermic machine perfusion: study protocol for an open-label, non-randomised, prospective, single-arm trial

Richard W Laing,[1,2] Hynek Mergental,[1,2] Christina Yap,[3] Amanda Kirkham,[3] Manpreet Whilku,[3] Darren Barton,[3] Stuart Curbishley,[2] Yuri L Boteon,[1,2] Desley A Neil,[1] Stefan G Hübscher,[1,2] M Thamara P R Perera,[1,2] Paolo Muiesan,[1,2] John Isaac,[1] Keith J Roberts,[1] Hentie Cilliers,[1] Simon C Afford,[2] Darius F Mirza[1,2]

For numbered affiliations see end of article.

**Correspondence to**
Professor Darius F Mirza;
darius.mirza@uhb.nhs.uk

## ABSTRACT

**Introduction** The use of marginal or extended criteria donor livers is increasing. These organs carry a greater risk of initial dysfunction and early failure, as well as inferior long-term outcomes. As such, many are rejected due to a perceived risk of use and use varies widely between centres. Ex situ normothermic machine perfusion of the liver (NMP-L) may enable the safe transplantation of organs that meet defined objective criteria denoting their high-risk status and are currently being declined for use by all the UK transplant centres.

**Methods and analysis** Viability testing and transplantation of marginal livers is an open-label, non-randomised, prospective, single-arm trial designed to determine whether currently unused donor livers can be salvaged and safely transplanted with equivalent outcomes in terms of patient survival. The procured rejected livers must meet predefined criteria that objectively denote their marginal condition. The liver is subjected to NMP-L following a period of static cold storage. Organs metabolising lactate to ≤2.5 mmol/L within 4 hours of the perfusion commencing in combination with two or more of the following parameters—bile production, metabolism of glucose, a hepatic arterial flow rate ≥150 mL/min and a portal venous flow rate ≥500 mL/min, a pH ≥7.30 and/or maintain a homogeneous perfusion—will be considered viable and transplanted into a suitable consented recipient. The coprimary outcome measures are the success rate of NMP-L to produce a transplantable organ and 90-day patient post-transplant survival.

**Ethics and dissemination** The protocol was approved by the National Research Ethics Service (London—Dulwich Research Ethics Committee, 16/LO/1056), the Medicines and Healthcare Products Regulatory Agency and is endorsed by the National Health Service Blood and Transplant Research, Innovation and Novel Technologies Advisory Group. The findings of this trial will be disseminated through national and international presentations and peer-reviewed publications.

> **Strengths and limitations of this study**
>
> ► The study will answer the question: 'Can ex situ end ischaemic normothermic machine perfusion safely increase the number of transplantable livers?'
> ► The study aims to establish objective liver viability criteria and biomarkers that may enable point-of-care assessment of liver quality.
> ► The study has clearly defined criteria characterising the discarded organs.
> ► Incorporation of an adaptive three-stage trial design provides opportunities to assess patients' safety, allowing for early trial termination if necessary.
> ► The trial includes low and moderate risk recipients only—the suitability for high-risk recipients will require further testing.

**Trial registration number** NCT02740608; Pre-results.

## INTRODUCTION

### Liver transplantation

Liver transplantation is a highly successful treatment for end-stage liver disease, fulminant hepatic failure and early-stage primary liver cancer. Deaths from liver disease have soared by 40% in a decade and continue to rise. Liver disease kills 11 000 a year in England and the average age of death from liver disease (59 years) continues to decrease.[1] Over the past 50 years, transplant techniques and outcomes have greatly improved and 5-year survival rates of 70%–80% mean that transplantation has become the mainstay of treatment for an increasing number of patients with chronic liver disease, metabolic

disorders, acute liver failure and malignancy. As such, the demand for donor livers greatly exceeds supply and approximately 20% of patients die while awaiting transplantation.[2] In Europe, the most common indications for liver transplantation are cirrhosis (68%), malignancy (14%) and acute hepatic failure (8%). The main causes for cirrhosis in Europe are the hepatotropic viruses and alcohol-related liver disease.[3] Non-alcoholic fatty liver disease is an emergent cause and despite health campaigns, the incidence continues to rise. In the UK, it is predicted that the incidences of end-stage liver disease and hepatocellular carcinoma will increase substantially during the next decade, exacerbating the existing shortage of donor livers.

### The UK liver transplant programme

Between March 2015 and April 2016, there were 1161 new waiting list registrations in the UK, and 878 transplants were carried out. Of the 621 patients on the list as of April 2015, 22% died or were removed from the list (n=135) due to deteriorating health.[4] This is reflected across other countries to the extent that a patient is now more likely to die within the first 12 months of being listed than the first 12 months' post-transplant.[5] Over the past decade, there has been a very modest increase in the use of standard or 'ideal' organ donors (those retrieved from young donors following a diagnosis of brain stem death, (DBD)). In response, centres have use donors following circulatory death (DCD) and suboptimal 'marginal' or 'extended criteria' donors (those of older age, livers with a presence of steatosis, etc).

### Responding to the shortage

There are several ways to respond to the shortage. Organ donation policies are undergoing changes; however, there is a lack of well-controlled scientific evidence on which to base decisions regarding policy-making and opinions are strong and divided. Spain has the highest organ donation rates and operates an opt-out system, however, the rise in rates only started approximately 10 years after the system's introduction. Wales is the most recent country to go down this route, however unlike in Spain, next of kin consent is still required before patients can become organ donors. More likely, the increased Spanish donation rates are due to a combination of factors—the creation of a transplant coordination network that operates at hospital, regional and national levels, the placement of transplant coordinators at each procurement hospital and the improvement in the quality of information received by the public. Living donation is one potential means to increase the number of liver transplants, using surgical techniques developed for liver resection and 'liver splitting' (which uses a single liver for transplantation into two recipients). The major limitations are most patients do not have a willing or suitable living donor and there are concerns about the risks to the healthy donor. The reported risk of donor death is estimated at 0.2% but the risk of serious complications is much higher.[6 7] Although programmes have had some success in countries without deceased donor programmes, living donor transplantation will be unlikely to have a significant impact on the shortage of donor livers in most countries.

### The use of 'marginal' or 'extended criteria' donors

As discussed, a rising proportion of transplants are carried out using 'marginal' or 'extended criteria' grafts, procured from obese or elderly donors with multiple comorbidities.[8] These livers are significantly more susceptible to cold storage-related ischaemic injury, which increases the risk of graft failure and recipient morbidity and mortality. Reflecting the issues with these suboptimal grafts, in 2014/2015, of 1282 solid organ donors, only 924 (72.1%) livers were deemed suitable for retrieval and only 812 (63.3%) were subsequently transplanted.[9] The duration of the functional warm ischaemic time (FWIT) is an important determinant of outcome. The recent document 'Donation After Circulatory Death' published by a steering group on behalf of the British Transplantation Society and Intensive Care Society suggested that the stand-down time from the onset of functional warm ischaemia for DCD liver transplantation was 30 min (although 20 min is ideal), and that age was an important factor. Because of this, a number of livers will be retrieved from DCDs that fall into the 'marginal donor' category and may not go on to be transplanted.[10]

Several donor parameters have been identified as relative risk factors for poor outcome including age, steatosis, DCD donation, split livers and prolonged cold ischaemia time (>12 hours). These were all developed using North American data and formulated into an algorithm known as the Donor Risk Index (DRI) and later validated using European data.[11 12] The British Transplantation Society have published their own guidelines on the use of donor organs and use criteria in table 1 to distinguish between grafts of varying quality.

### Organ preservation

The current standard of donor liver preservation is based on static cold storage (SCS).[13] During SCS, organs are flushed and cooled with specific chilled preservation solutions (University of Wisconsin solution is used most commonly although histidine–tryptophan–ketoglutarate (HTK) solution is also used less widely) and ice is added to the abdominal cavity. After retrieval, the organ is placed in fluid-filled sterile plastic bags for transportation and stored in preservation solution within an ice box until transplantation. Although the available preservation solutions differ in chemical composition, their function is essentially the same. The hypothermia aims to reduce the liver's metabolic activity and the solution aims to reduce the cellular swelling. This is a consequence of anaerobic metabolism resulting in depletion of ATP stores leading to influx of free calcium and activation of phospholipases.[14] Cooling the organ slows metabolism approximately 12-fold but cannot prevent its dysfunction and the eventual destruction of cellular integrity.

**Table 1** Criteria for donor quality as per British Transplantation Society UK Guidelines for donors after circulatory death

| Good livers—all should be used (DBDs and DCDs) | Ideal livers—all should be used (DCDs) | Marginal donors—use selectively (DCDs) | Absolute contraindications to using liver as donor organ |
|---|---|---|---|
| ▶ Age<50 | ▶ Age<50 years | ▶ Age >50 years | ▶ DCD with macrovesicular steatosis >30% |
| ▶ Normal LFTs | ▶ Weight<100 kg | ▶ Weight >100 kg | ▶ ESLD |
| ▶ <5 days on ICU | ▶ FWIT<20 min | ▶ FWIT 20–30 min | ▶ Acute liver failure |
| ▶ Low levels of inotropic support | ▶ CIT<8 hours | ▶ CIT 8–12 hours | ▶ Acute liver injury that is not improving |
| ▶ <30% steatosis | ▶ <15% steatosis | ▶ >15% steatosis | |
| ▶ No active sepsis | ▶ ICU stay<5 days | ▶ ICU stay >5 days | |

CIT, cold ischaemic time; DBD, donor following brain death; DCD, donor following circulatory death; ESLD, end-stage liver disease; FWIT, functional warm ischaemic time; ICU, intensive care stay; LFTs, liver function tests.

Ischaemia reperfusion is an important factor influencing graft outcome.[15] The ischaemic phase starts early in the procurement process (swings in blood pressure following brain death or due to the functional warm ischaemic time in non-heart beating donors) and triggers a complex cascade of cellular and molecular events including the release of proinflammatory mediators and chemotaxis of cell types that initiate progressive immunological processes. During the reperfusion phase, 'the reflow paradox' causes infiltration of the tissues by leucocytes and cellular injury occurs through a series of pathways that include lipid peroxidation and the creation of reactive oxygen species[16] The most common manifestation of the ischaemia-reperfusion process is delayed graft function, which is the inability of the organ to fulfil the physiological needs of the recipient and is associated with graft failure, retransplantation and death.[17] Static cold storage therefore is unable to reverse the injury sustained during donor death and procurement, causes injury due to the cooling process, limits the preservation time and prevents physiological assessment prior to transplantation.

### In situ organ reconditioning

To reverse or diminish the injury, many cytoprotective strategies have been tested in experimental models of transplantation and several have been shown to have therapeutic potential, including gene therapy,[18 19] cytokine or growth factor administration,[20–22] vasodilating agents and ischaemic preconditioning.[23 24] Treatment of the organ during preservation has major logistic and ethical advantages over any attempt to achieve the same effects by treating the donor (therapeutic interventions before declaration of death are not currently permitted unless they are of potential benefit to the donor). Recently, there has been published early experience with normothermic regional perfusion of DCD donors, nevertheless the feasibility and benefit of this experimental approach is yet to be shown.[25]

### Normothermic machine perfusion of the liver

Bretschneider and Starzl first attempted machine perfusion of the liver in the late 1960s. Although hypothermic machine perfusion (HMP) has been shown some promise in clinical studies, normothermic machine perfusion of the liver (NMP-L) combats the limitations of SCS previously described by aiming to maintain the organ at the body's natural temperature while providing oxygen, nutrition and the essential substrates necessary for adequate cellular metabolism. Providing a homeostatic environment theoretically enables us to extend our storage period and test the organs physiological parameters. To date, only one clinical trial of 20 adult recipients of livers maintained by HMP has been published showing a reduction in early graft dysfunction (5% vs 25% p<0.08) as well as a significant reduction in serum injury markers in the HMP group. A joint pilot trial between Oxford University, King's College Hospital London and University Hospitals Birmingham Foundation Trust (UHBFT) recruited 20 patients into an NMP-L phase I study and concluded the procedure was feasible and safe when used on current conventional donor acceptance criteria.[26] Following this, a 220-patient phase III international clinical trial entitled 'consortium for organ preservation in Europe (COPE) work package 2 (WP2)' has completed recruitment and the results are eagerly awaited. The Liver Unit at University Hospital Birmingham NHS Foundation Trust contributed to this multicentre international trial by randomising 50% of the study patients.

Our group believes NMP-L enables the donor organ to be functionally assessed, thereby increasing transplant safety. It can also extend organ preservation times to improve transplant logistics and donor organ use. There are several devices available on the market, but only the OrganOx metra has been widely used in the clinical transplant setting.[26] Our team has performed over 70 liver transplants with grafts preserved on this machine and has gained broad experience by using this device. The OrganOx metra is the leading device in terms of the number of clinical transplants undertaken, with more than 100 machine-perfused livers transplanted in the phase III randomised European trial, together with 20 livers in the phase I safety study and further ongoing trials in North America. For these reasons, we have decided to use the OrganOx metra device for the proposed study.

The device consists of a unit that cradles the liver, a perfusate reservoir, oxygenators, pumps operating at physiological pressures and a closed tubing system that connects the unit to the portal vein, hepatic artery and vena cava. The constituents of the perfusate can vary but

generally consist of whole blood for oxygen carriage, sources of nutrition (glucose, insulin, amino acids), anti-thrombotic agents (heparin, epoprostenol), antibiotics and acid-base agents which help reduce cellular oedema, cholestasis, microvascular injury and the effects of free radicals.

## Benefits of NMP-L

NMP-L does not simply benefit marginal DCD organs that have been exposed to a damaging FWIT. DBD is a catastrophic physiological event associated with profound hypotension (parasympathetic response) followed by hypertension, tachycardia and high levels of circulating catecholamines (sympathetic surge) followed by another reduction in the sympathetic outflow. These dramatic swings can cause significant graft ischaemia prior to retrieval. Diabetes insipidus occurs in 70%–80% of brain dead patients causing severe hypernatraemia (associated with primary liver graft non-function), hypokalaemia, hypocalcaemia, hypophosphataemia and hypomagnesaemia.[27 28] Pirenne et al described seven cases when livers from DBDs between 70 and 80 years old were used with 'favourable outcomes'.[29] More recently, groups from Italy have reported excellent outcomes using grafts from octogenarian donors.[30 31] NMP-L could however play an important role in preconditioning and assessing such organs prior to transplantation.

Hellinger et al were unable to identify a benefit using NMP-L in 1997; however, it was the first study of its kind.[32] In 2001, Schon used NMP-L to preserve and recondition livers that had been exposed to 1 hour of warm ischaemia. These livers were then transplanted into pigs which all survived longer than 7 days. The group that received livers preserved using SCS had no survivors.[33] Several studies have been published by the Oxford group, responsible for OrganOx metra. Imber et al published results from a study on a porcine model comparing NMP-L with SCS controls. They showed livers preserved using NMP-L were significantly superior (p<0.05) to SCS livers 'in terms of bile production, factor V production, glucose metabolism and galactose clearance', while SCS livers had higher perfusate levels of hepatocellular enzymes and more cellular damage.[34] The same year, they successfully perfused and maintained five porcine livers for 72 hours, managing to maintain normal physiological parameters, pH, protein synthesis and histological architecture.[35] In 2009, Brockman et al simulated DBD and DCD scenarios in a porcine model. After 5 hours of preservation (NMP-L vs SCS), there was no difference seen in preservation method in either the DCD or DBD graft recipients. After 20 hours of preservation, however, both DCD and DBD grafts that had been preserved using NMP-L were superior to their SCS counterparts with respect to enzyme release, histological changes and recipient survival. Of note, there was no difference in survival between DCD and DBD NMP-L-preserved graft recipients (83% and 86%, respectively).[36]

## Preclinical research and pilot study

Our team's preclinical research on rejected human livers has demonstrated that metabolism of lactate, in combination with bile production, maintenance of physiological pH and stable blood flow rates are sensitive parameters predictive of organ viability. In April 2014, the UHBFT Novel Therapeutics Committee approved a pilot clinical project for transplantation of five reconditioned liver grafts, initially deemed unusable for transplantation. In this series, livers were declined by all the UK transplant units, after which NMP-L commenced following a variable period of SCS. Still, five out of six tested livers met the viability criteria and were successfully transplanted.[37] Although this pilot project showed that viability testing has the potential to transform the organ selection and acceptance process of high-risk livers, our observation primarily provided the feasibility and short-term outcome data. In addition, this cohort also demonstrated the feasibility of performing NMP-L within a 'back-to-base' model, that is, following SCS and inspection at the transplant centre. This offers logistical and financial advantages over using NMP-L in place of SCS and may target livers that would benefit the most from NMP-L. More research in this area is required, and this was recognised by the Health Innovation Challenge Committee of the Wellcome Trust who awarded our study group a research grant to fund this trial. We have demonstrated so far that a proportion of currently rejected liver allografts might be salvaged by subjecting them to NMP-L and viability testing. Use of this technology could transform the use of high-risk organs and may improve access to treatment for thousands of patients awaiting liver transplantation globally.

## METHODS
### Study design overview

Viability testing and transplantation of marginal livers (VITTAL) is an open-label, non-randomised, prospective, single-arm trial, using NMP-L testing viability and transplantation of marginal livers. It is being conducted at a single site (UHBFT). The design uses two linked components assessing: (1) the feasibility of NMP-L as a technique to increase the number of transplantable livers and (2) achievement of successful transplantation of the NMP-L treated marginal livers. (1) uses a two-stage adaptive design,[38] requiring up to 53 marginal livers to be perfused. (2) uses a three-stage adaptive design[39] and requires 22 NMP-L treated marginal livers to be transplanted. Success is measured by a 90-day patient survival—a nationally accepted, monitored and continually audited outcome following liver transplantation.

### Ethical and regulatory approval

The National Research Ethics Service (NRES) London-Dulwich (REC reference 16/LO/1056, Protocol number RG 15–240) and the Medicines and Healthcare Products Regulatory Agency (MHRA) approved all versions of the study protocol. This trial will use the

OrganOx metra device following a variable period of SCS to evaluate organ viability pretransplant procedure. The OrganOx metra device currently has a Conformité Européene (CE) mark for liver organ transport and not organ evaluation. The use of the device within this clinical trial is therefore off registration and UK Competent authority (MHRA) clinical trial Investigation: No Objection was obtained (MHRA ref: CI/2016/0031. In addition, approval from the Research and Development (R&D) department at UHBFT and from National Health Service Blood and Transfusion (NHSBT's) Research, Innovation and Novel Technologies Advisory Group (RINTAG) was obtained prior to the start of screening.

### Graft entry into study and subsequent preparation

The patient and donor liver pathways can be seen in figure 1. All livers will be retrieved with the intention and standardised technique to use them for transplantation. Following the retrieval procedure at the donor hospital the liver will be placed in ice-cold preservation solution on the back table and transported (according to local protocol). If the liver is allocated to UHBFT, if it is then considered not suitable for use it must be rejected by the on-call transplanting surgeon. For the liver to be considered untransplantable, the liver will be inspected by the on-call transplant surgeon and another transplant surgeon in the department. The liver will

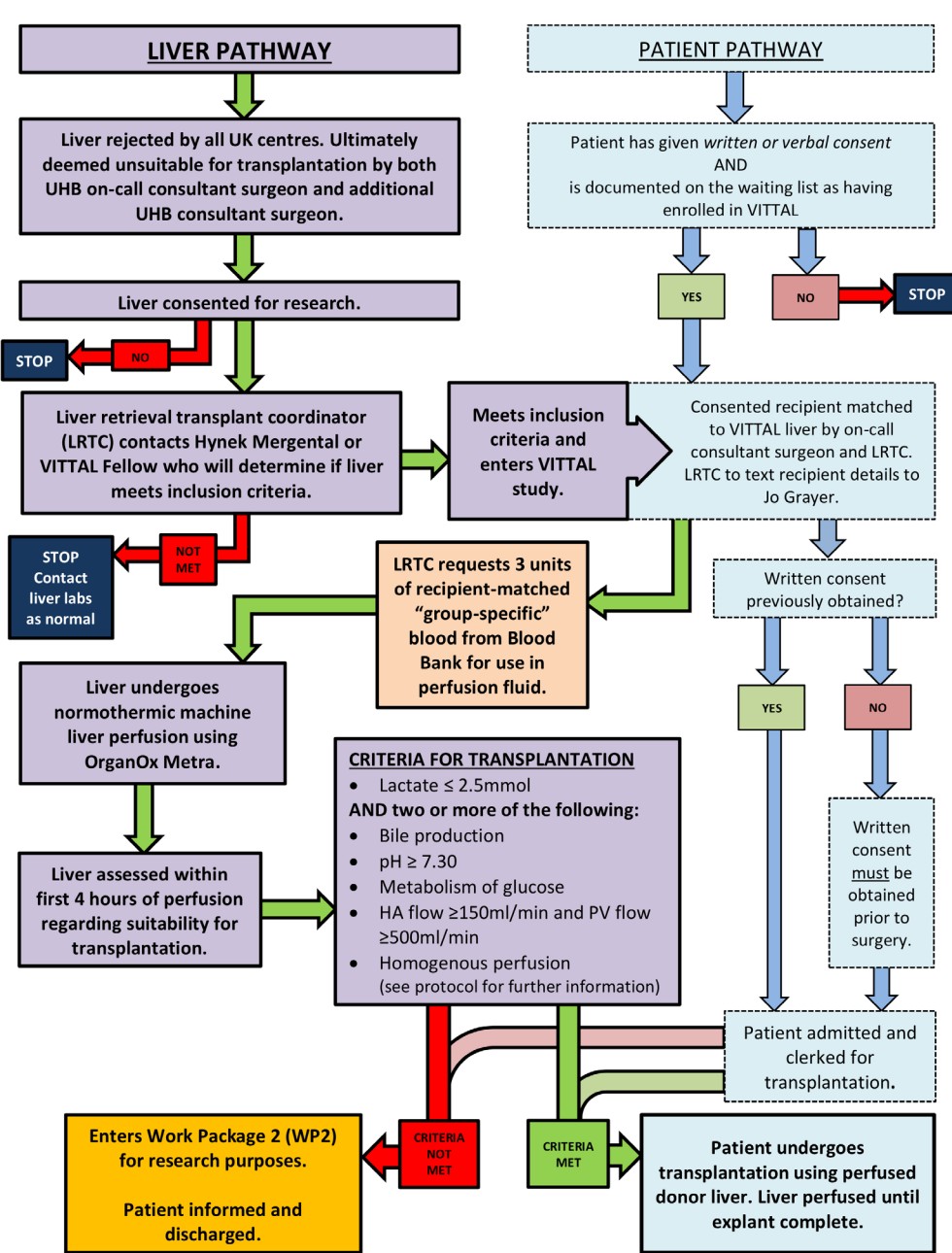

**Figure 1** Patient and donor liver pathways. HA, hepatic artery; PV, portal vein; UHB, University Hospitals Birmingham; VITTAL, Viability testing and transplantation of marginal livers.

then be offered as a fast track graft to the other centres around the UK. If rejected by all centres and if consent for research was taken, it will be considered for use in VITTAL. Livers offered to our unit as fast track offers from other centres will undergo the same two-consultant rejection process. An appropriate consented potential recipient will be selected by the transplant surgeon and contacted by the coordinator and will come into hospital for admission. The coordinator will request three units of packed red blood cells, matched to the intended recipient, for use in the OrganOx metra device. The liver will be prepared according to the procedure for preparing the device for use and placing the organ on the device (described in detail in the OrganOx metra Instructions for Use (IFU) document (V.13.0, 12 March 2016). The liver will be weighed prior to being connected to the device. If cannulation proves impossible, the liver will be rejected as previously intended. If the liver meets the criteria for transplantation, the recipient explantation will commence and the procedure for removing the liver from the device is also described in the IFU. Implantation and reperfusion of the liver will proceed as per the usual practice of the implanting centre. The patient will be clerked as if they were being admitted for a standard liver transplant.

## Perfusion of the graft

The machine will be primed with a perfusate suitable for NMP-L and will use packed red cells as the oxygen carrier. During the perfusion, biochemical analysis of the blood-based perfusate will be performed using a Cobas biochemical point-of-care analyser (Roche Diagnostics) which will give results for pH, $pO_2$, $pCO_2$, bicarb, base excess, calcium, chloride, sodium, potassium, haemoglobin, haematocrit, lactate and glucose. Arterial and portal venous flows, resistances and pressures will also be recorded. Samples to be collected are detailed in table 2.

The duration of machine perfusion will be dictated by logistics and the recipient's explant but should not be less than 4 hours or more than 24 hours. For a graft to be considered for transplantation it must *meet at least 2 of the following criteria* within 4 hours of the start of perfusion:

► Metabolise lactate to less than or equal to 2.5 mmol/L within 4 hours of the start of the perfusion;
► Demonstrate evidence of bile production;
► Maintain a pH greater than 7.30;
► Show evidence of glucose metabolism;
► Maintain stable hepatic arterial flow of more than or equal to 150 mL/min and portal flow more than or equal to 500 mL/min;
► Achieve homogeneous graft perfusion with soft consistency of the parenchyma.

Once the transplanting surgeon is content that the liver has met the criteria required for transplantation, the recipient will be brought to theatre and the explant will commence.

Explantation, implantation and reperfusion of the liver will be carried out in using standardised techniques by

| Table 2 | Trial sample collection schedule | | |
|---|---|---|---|
| Perfusate samples | Hepatic arterial and hepatic venous biochemistry (point of care) | Preperfusion Every 30 min during perfusion | Cobas point-of-care desktop analyser |
| | Perfusate supernatant | Preperfusion Every 15 min for first hour Every hour thereafter | 5×1 mL aliquots Stored at −80°C |
| Liver samples | Liver biopsy | L1 Preperfusion L2 After 4 hours L3 at end of perfusion* L4 Postreperfusion | 16 G core needle biopsy Divided into segment for formalin, segment for frozen and piece for electron microscopy |
| | Common bile duct | CBD1 Preperfusion CBD2 Postreperfusion | Formalin |
| Bile samples | (If produced) | B2 sample at 2 hours B4 sample at 4 hours B6 sample at 6 hours | Total volume recorded and 2 mL samples snap frozen at these time-points |
| Patient samples | Biochemistry haematology clotting | Visits 1, 2, 3, 4, extended follow-up | Standard of care |
| | Serum, plasma, mononuclear cells (PBMC) | Visit 1 (preoperative (postinduction of anaesthesia), postreperfusion day 4 postoperative) Visits 2, 3, 4, | Additional research samples |
| | Urine | Visit 1 (pre-operative [post-induction], postreperfusion Day four post-op) | Additional research samples |

*If lasting longer than 6 hours.
PBMC, peripheral blood mononuclear cell.

   Laing RW, *et al*. *BMJ Open* 2017;7:e017733. doi:10.1136/bmjopen-2017-017733

the on-call transplant surgeon. The liver will remain on the machine until after the explantation has taken place at which point it will be flushed by 2 L of cold HTK immediately prior to implantation.

## Concomitant therapy/medications

Patients will receive immunosuppression according to hospital protocols and other medications as necessary for their comorbidities and current clinical condition. Their postoperative care will be the same as if they had undergone a standard liver transplant.

## OBJECTIVES AND OUTCOME MEASURES
### Primary

There are two linked primary objectives and respective outcome measures:

Primary objective: (A) Establish the feasibility of NMP-L to increase the number of transplantable livers.

Primary Outcome measure: (A) 'Rescue rate' that is, the proportion of rejected livers that can be used for transplantation having been deemed viable following a period of machine perfusion.

Primary objective: (B) Achieve successful transplantation of previously rejected donor livers following viability testing using NMP-L.

Primary outcome measure: (B) 90-day patient survival, calculated as the number of patients alive 90-day post-NMP-L-treated marginal liver transplantation (numerator) divided by the total number of NMP-L-treated marginal liver transplants performed (denominator).

### Secondary

Secondary objective (1): Assessment of liver graft function following transplantation (by incidence of primary non-function and early allograft dysfunction).

Secondary outcome measures (1): Liver function tests; 90-day graft survival; 12-month patient and graft survival.

Secondary objective (2): Assess morbidity associated with receipt of extended criteria graft that had previously been rejected.

Secondary outcome measures (2): Adverse event rates and severity, graded according to the Clavien-Dindo classification[40] (see online supplementary file 1); Requirement of renal replacement therapy; incidence of biliary complications (including incidence of ischaemic-type biliary lesions diagnosed on magnetic resonance cholangiopancreatography (MRCP) at 6 months); incidence of vascular complications; biopsy-proven acute rejection; reoperation rate; length of intensive therapy unit stay and length of hospital stay.

Secondary objective (3): Assess the physiological response to reperfusion of the perfused grafts

Secondary outcome measures (3): Postreperfusion syndrome, defined as a decrease in mean arterial pressure (MAP) of more than 30% from the baseline value for more than 1 min during the first 5 min after reperfusion (assessed in the context of inotrope use).

Secondary objective (4): Identify impact on quality of life (QoL) after transplantation with these liver grafts.

Secondary outcome measures (4): Quality of life by delivery of the EuroQoL 5 Dimensions 5 Levels questionnaire at baseline, day 30 and 6 months post-transplant.

## ANALYTICAL METHODS
### Histopathology

Two independent liver histopathologists from UHBFT will perform all the histopathological assessments. Both will be blinded to the graft type and the primary and secondary outcome measures although the presence or absence of a postreperfusion biopsy means they will know whether a graft has met the criteria for transplantation. The histological analysis will be established using H&E at two levels as well as periodic acid Schiff, periodic acid Schiff diastase, haematoxylin van Gieson, reticulin, orcein, rhodanin and Perls stains of formalin-fixed paraffin-embedded liver tissue.

### Perfusion, clinical and laboratory data

Donor and patient demographics as well as intraoperative data will be collected. Body mass index (BMI) was defined as weight in kilograms divided by the square of the height in metres ($kg/m^2$). In non-heart beating (DCD) donation, FWIT is defined as the time between the systolic blood pressure of the donor dropping below 50 mm Hg until the point of aortic perfusion. Cold ischaemic time is defined as the time between aortic perfusion and the start of NMP-L. DRI and Balance of Risk (BAR) will be calculated as per the relevant literature.[11 41]

The perfusate fluid will undergo point-of-care biochemical testing every 30 min as previously described. Perfusate will be taken at the time-points described in table 2 and tested for transaminase, urea, albumin and factor V levels. Patient's blood samples will be analysed for full blood count, urea, creatinine and electrolytes, liver function tests, prothrombin time, amylase, C-reactive protein and plasma glucose using standard laboratory methods (Roche Modular system, Roche, Lewes, UK) both preoperatively and postoperatively. Research recipient blood and urine samples will also be taken as part of WP2 that will enable immune cell profiling as well as lipodomic, proteomic and metabolomic testing.

### Patient questionnaires

QoL will be assessed by delivery of the EQ-5D-5L questionnaire (UK (English) 2009 EuroQol Group EQ-5D is a registered trademark of the EuroQol Group) at baseline, day 30 and 6 months' post-transplant. EQ-5D-5L is a five-level version of the EQ-5D descriptive system (Herdman *et al.* Qual Life Res DOI 10.1007/s11136-011-99031). The 5L retains the 5-dimensional (5D) structure of the original EQ-5D-3L but the levels on each dimension were expanded to five based on qualitative and quantitative studies conducted by the EuroQol Group. Index-based values ('utilities') enable the calculation of quality-adjusted life years which help inform economic evaluations of healthcare interventions.

## STATISTICAL JUSTIFICATION AND OUTCOME ANALYSIS
### Sample size justification

For (A) feasibility of NMP-L to rescue discarded liver grafts, it is anticipated that NMP-L will achieve a desirable organ recovery rate of at least 50%, with an undesirable rate of 30% or less as this would not be considered economically feasible. The significance level ($\alpha$) is set at 0.05, corresponding to the probability of incorrectly rejecting the hypothesis given it is true (type I error), and the power is set at 0.90 (type II error rate, $\beta = 0.10$), corresponding to the probability of correctly deciding the NMP-L treatment is successful given the true response rate is greater than 50%.

Using a Simon's two-stage design[38]:

Interim assessment stage 1A of accrual: 24 marginal grafts will be perfused and assessed in the first stage. Grafts will be transplanted depending on the criteria achieved. The procedure will be considered infeasible if there are fewer than eight recovered livers. If more than eight livers are transplanted, we will proceed to Stage 2A.

Final stage 2A of accrual: Up to additional 29 marginal grafts will be perfused. We would consider the procedure feasible if there are at least 22 recovered livers out of 53 perfused livers.

For (B) for viable livers transplanted following NMP-L, a desirable 90-day patient survival rate is at least 88%, with an undesirable rate of 73% (15% lower). The mean 90-day patient survival rate for 'standard' liver transplants is 93%.[42] An optimal three-stage design[39] will be used to test the null hypothesis that the mean 90-day patient survival rate will be less than 73% (p≤0.73) versus an alternative hypothesis that the 90-day patient survival rate will be at least 88% (p≥0.88). The significance level is set at 0.20 (target α=0.2), giving a 0.2 probability to conclude that a single transplantation is viable when it truly is not viable. The power is set at 80% (target β=0.2), giving a 0.2 probability to conclude that a single transplantation is not viable when it truly is viable.

Interim assessment stage 1B: Following transplantation in three patients, the trial will stop early (concluding p≤0.73) if there are fewer than two patients achieving 90-day survival. If two or more patients reach the primary end point of 90-day survival, an additional eight transplantations will be performed.

Interim assessment stage 2B: Following transplantation in 11 patients (combined first and second stages) the trial will stop early (concluding p≤0.73) if there are 7 or fewer successes. If 8 or more patients reach the primary end point, an additional 11 transplantations will be performed.

Final stage 3B: Following transplantation in 22 patients in all three stages, the trial will be successful if at least 18 patients reach the primary end-point of 90-day survival.

The trial schema is provided in figure 2.

### Analysis of outcome measures
#### Primary analysis

To assess (A) the feasibility of NMP-L, the rescue rate will be calculated as the number of perfused marginal grafts meeting the criteria for viability (numerator) divided by the total number of perfused marginal grafts (denominator).

$$Rescue\ Rate = \frac{Number\ of\ viable\ perfused\ marginal\ grafts}{Total\ number\ of\ perfused\ marginal\ grafts}$$

To assess (B)—achievement of successful transplantation of previously rejected donor liver following viability testing using NMP-L. We will evaluate 90-day patient survival rate as an indicator of liver function and/or viability following transplantation of marginal liver grafts following NMP-L. The 90-day patient survival rate will be calculated as the number of patients alive at 90-day post-transplant with a VITTAL graft, divided by the total number VITTAL patients transplanted.

$$90\ day\ patient\ survival\ rate = \frac{Number\ of\ patients\ alive\ at\ 90\ days\ post\ transplantation}{Total\ number\ of\ transplanations\ performed}$$

For (A), all livers undergoing NMP-L treatment will be included for evaluation in the interim and final analyses. For (B), all transplantations performed will be included for evaluation in the interim and final analyses. The rate outcomes will be reported together with confidence intervals using the Wilson (1927) method.[43]

#### Planned interim assessments

As we have used adaptive designs, there are planned formal interim assessments for both (A) feasibility of NMP-L and (B) successful transplantation of rescued livers, with clear 'Go'/'No go' decisions as detailed earlier. Ideally, recruitment (ie, transplantation) would stop while interim analyses of the primary outcome measures are performed. For (A), this could happen immediately, however for (B), this would result in a pause of over 3 months hence the pragmatic approach for such adaptive designs is to continue recruitment while they are being conducted.

To maximise patient safety, for (B) at the end of the first stage (transplantation of the first three patients), recruitment will be paused to allow the Data Monitoring Committee (DMC) to assess the initial safety data. Once all three patients are discharged, if the DMC considers the patients to be recovering well, with liver function that would be expected at this stage, recruitment can continue prior to the patients reaching the primary endpoint of 90-day survival. A follow-on report will be sent to the DMC once the third patient reaches the primary end-point. For the second stage (transplantation of 11 patients), safety data will be sent to the DMC for review after discharge of all 11 patients however recruitment need not stop at this point. A follow-on report will again be sent once the 11th patient reaches the primary end-point.

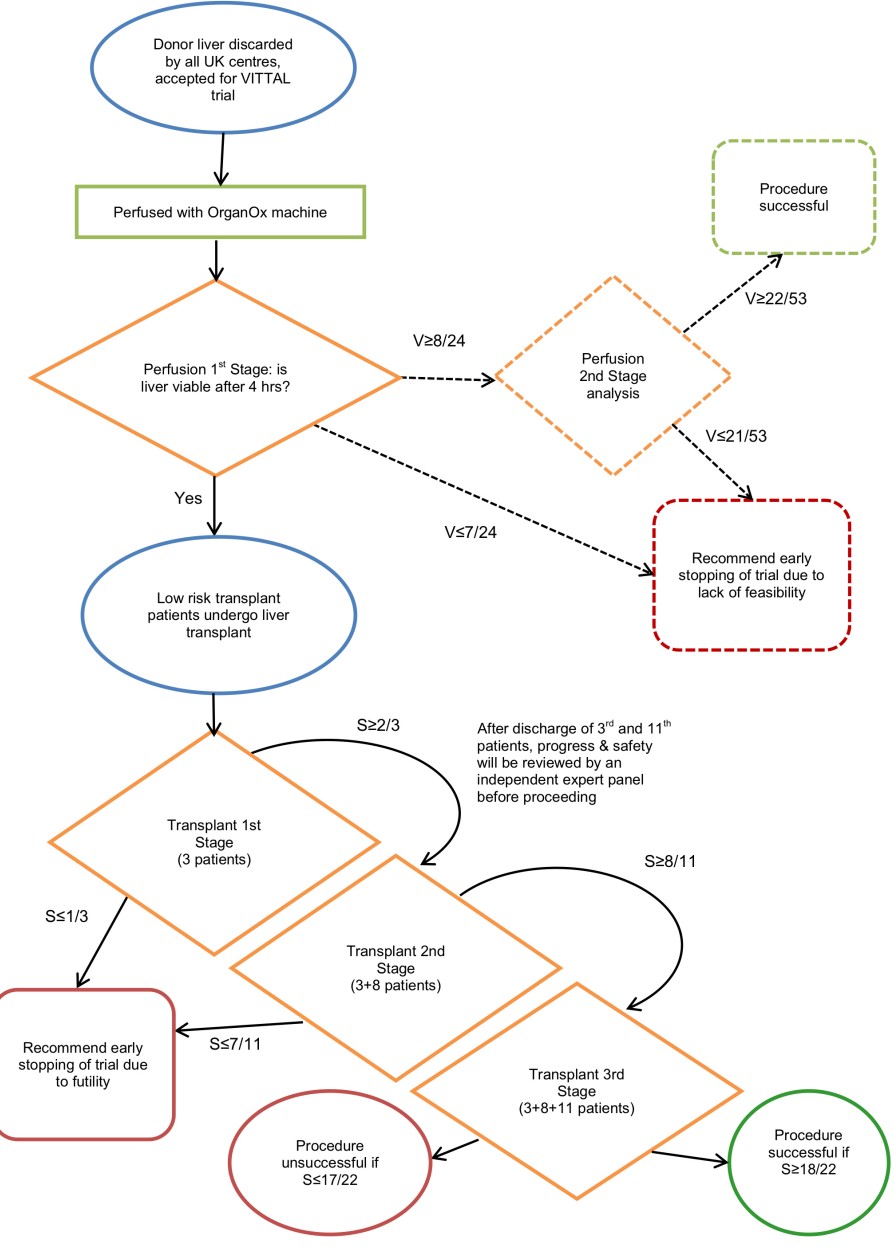

**Figure 2** Trial schema. VITTAL, Viability testing and transplantation of marginal livers.

Additional DMC meetings will be conducted on request if the success criteria are not met. If recruitment is fast, prompt reviews will be necessary to ensure the use of interim decisions.

## Secondary analysis

For all secondary outcome measures, analyses will be mainly descriptive. Continuous exploratory measures will be summarised via means, medians, SD and ranges. Categorical measures will be summarised with number and proportion in each category. To model repeated measures over time (eg, QoL), a linear mixed effects model (considering subject correlation) using parametric and more flexible models may be considered. Time-to-event outcomes

will be assessed using the method of Kaplan and Meier. Median survival with corresponding 95% CI will also be reported where appropriate. The assessment of graft function post-transplantation by incidence of primary non-function and early allograft dysfunction will be carried out by comparing results with a contemporary matched recipient group of patients obtained from a prospectively maintained database, with adjustment for potential confounders.

The contemporary matched recipient group will be matched using the following:

► Patient characteristics: age, sex, BMI, model for end-stage liver disease (MELD), UK end-stage liver disease (UKELD), aetiology;
► Donor liver characteristics: DCD or DBD, sex.

## Conduct of trial

### Donor liver selection

Suitable donor liver grafts will be selected from October 2016. Grafts will be retrieved with the intention to transplant and rejected as previously described.

### Graft inclusion criteria

Rejected donor liver grafts must meet all of the following inclusion criteria to be eligible for inclusion in the VITTAL trial:

► Liver from a donor primarily accepted with the intention for clinical transplantation;
► Rejected by all the other UK transplant centres via normal or fast-track sequence;
► Cold ischaemic time less than 16 hours for DBD and 10 hours for DCD grafts;
► One of the following parameters which would denote the marginal condition of the liver:
  – Donor risk index greater than 2.0[11];
  – Graft macrovesicular steatosis greater than 30%;
  – BAR score greater than 9[44];
  – Donor warm ischaemic time greater than 30 min;
  – Anticipated cold ischaemic time greater than 12 hours for DBD or 8 hours for DCD liver grafts;
  – Suboptimal liver graft perfusion documented by a photo of macroscopic appearance;
  – Donor transaminases (aspartate transaminase (ALT) or alanine transaminase (AST)) above 1000 IU/mL.

### Graft exclusion criteria

Livers meeting any of the following criteria would not be suitable for the VITTAL trial:

► Grafts from patients with active Hepatitis B, C or HIV infection;
► Livers with macroscopic appearance consistent with cirrhosis;
► Livers with advanced fibrosis;
► DCD grafts with donor warm ischaemic time (systolic blood pressure less than 50 mm Hg to aortic perfusion) more than 60 min;
► Excessive cold ischaemic times (DBD more than 16 hours / DCD more than 10 hours);
► Paediatric donor (<18 years);
► ABO (blood group) incompatibility.

### Recipient inclusion criteria

Suitable potential VITTAL graft recipients will be identified during the listing process. Patients will be told that they are potentially suitable to receive a graft from the VITTAL trial and will be given the patient information sheets to read more about the trial. If already listed, potential recipients will be identified on the list, contacted and sent the same documentation. If they wish to take part, a minimum of verbal consent will be taken. Enrolling in the trial will in no way impact on the chance of them receiving a standard 'transplantable' graft. Patient's with all aetiologies of chronic liver disease will

be considered for inclusion. Listed patients must meet all of the following inclusion criteria to be eligible for participation in the VITTAL trial:

► Adult primary liver transplant recipient;
► Patient listed electively for transplantation;
► Low to moderate transplant risk candidate, suitable for marginal graft, as assessed by the UHBFT liver transplant listing multidisciplinary team (MDT) meeting (these are usually candidates with low UKELD score, without cardiovascular comorbidities, with good functional and nutrition status, with patent portal vein and with no history of previous major upper abdominal surgery, for example, patients transplanted for liver cancer);
► There is no lower limit for MELD or UKELD. Upper UKELD is discussed in the exclusion criteria below.

### Recipient exclusion criteria

Subjects who meet any of the following exclusion criteria are excluded from participating in the VITTAL trial:

► 'High-risk patients' and recipients not considered suitable for a marginal graft (these are mainly patients with high UKELD score (>62 as per the NHSBT Liver Advisory Group (LAG) criteria for graft sharing in high-risk recipients in the North East of the UK with cardiovascular comorbidities or renal insufficiency, with poor nutrition and performance status or history of major upper abdominal surgery, for example, patients listed for liver retransplantation) (http://www.odt.nhs.uk,/search 'Liver Allocation Policy');
► Patients with complete portal vein thrombosis diagnosed prior to the transplantation;
► Liver retransplantation;
► Patients with fulminant hepatic failure;
► Patients undergoing transplantation of more than one organ;
► Contraindication to magnetic resonance imaging (ie, pacemaker fitted).

### Adverse events reporting and analysis

The collection and reporting of adverse events (AEs) will be in accordance with the Research Governance Framework for Health and Social Care and the requirements of the National Research Ethics Service (NRES). Definitions of different types of AEs are listed in online supplementary file 1. The reporting period for AEs will commence at visit 1 and end at the 24-month follow-up. The Investigator should assess the seriousness and causality (relatedness) of all AEs experienced by the patient (this should be documented in the source data) with reference to the protocol. This will include abnormal laboratory findings which are reported as clinically significant. All AEs, device deficiencies and adverse device event (ADEs) will be reported using the applicable electronic case report form (eCRF). AEs will be reported in accordance with Clavien-Dindo classification of surgical complications.[40] Anticipated AEs include those related to any form of major surgery; infection (chest, urine, blood, bile, wound, abdominal),

fluid collection (abdominal, pleural), renal dysfunction, cardiac failure, respiratory failure and those related to the disease process and transplantation; early allograft dysfunction, rejection, hospitalisation for pre-existing condition that has not deteriorated, clinically significant abnormal laboratory finding or other abnormal assessments that is associated with the condition being studied (unless judged by the investigator as more severe than expected for the patient's condition). The investigator will exercise his/her medical judgement in deciding whether an abnormal laboratory finding or other abnormal assessment is clinically significant. However, if in the opinion of the investigator, the frequency or severity of the event is greater than would be expected then it must be reported. Device deficiencies that did not lead to an adverse event but could have led to a medical occurrence if suitable action had not been taken or intervention had not been made or if circumstances had been less fortunate will also be recorded and reported.

## Those events not being reported

The following are considered routine during or after liver transplantation and will not be reported as AEs.

▶ Initial admission to intensive care following liver transplant;
▶ Elevation of AST and/or ALT<2000 iu/mL within 48 hours of liver transplant;
▶ Transfusion of ≤5 units of packed red cells;
▶ Transfusion ≤8 units of fresh frozen plasma;
▶ Transfusion ≤2 adult doses of platelets.

In addition to the above, medical and scientific judgement should be exercised in deciding whether expedited reporting is appropriate in other situations, such as important medical events that may not be immediately life threatening or result in death or hospitalisation but may jeopardise the patient or may require intervention. Any death occurring during the protocol defined follow-up period (within 90 days), whether considered device related or not, must be reported as an SAE within 24 hours of the local investigator becoming aware of the event. If a death occurs in a patient receiving a transplant, the cause of death will be investigated and reviewed by the Trial Management Group (TMG) and clinical team caring for the patient. Entry of patients in to the study would be temporarily suspended until these investigations are complete.

## Study visit overview

The VITTAL trial involves a minimum of four patient visits which all coincide with standard admissions either for surgery or for outpatient follow-up. There are no additional trial-specific visits. The schedule for the study visits and data collection is summarised in table 3. Visit 1 encapsulates admission for transplant and the postoperative period if the transplant proceeds. Visits 2, 3 and 4 are scheduled for 30-day, 90-day and 180-day follow-up, respectively. All patients will undergo MRCP during visit 4 to investigate the occurrence of ischaemic-type biliary lesions which also marks trial end-point. Patients will continue to be followed up at 12 months and 24 months as part of their standard post-transplant care, and data will be collected at these time-points for long-term reporting.

## Storage of samples

Patient blood samples taken as part of their standard of care will be processed and stored according to UHBFT procedures. Perfusate, patient serum, plasma, urine samples and mononuclear cell preparations collected during visits 1–4 will be stored frozen in 0.5–1.0 mL aliquots at −80°C at the Institute of Biomedical Research, University of Birmingham. Liver biopsy tissue specimens will be collected and the formalin fixed paraffin embedded segments will be processed by staff in the department of cellular pathology at UHBFT. After sectioning and staining, tissue blocks will be stored at the Institute of Biomedical Research. All samples will be collected in accordance with national regulations and requirements including standard operating procedures for logistics and infrastructure. Samples will be taken in appropriately licensed premises, stored and transported in accordance with the Human Tissue Authority guidelines and trust policies.

## Data handling, quality assurance, record keeping and retention

Data will be managed according to the standard operating procedures of the Cancer Research UK Clinical Trials Unit (CRCTU) at the University of Birmingham, UK. The CRCTU is fully compliant with the Data Protection Act 1998 and the Guidelines for Good Clinical Practice. The CRCTU will monitor the trial and provide annual reports to the MHRA. The trial is registered with the Data Protection Act website at the University of Birmingham. Donor and patient details will be kept anonymous (specific study identification codes will be used for each study donor). Anonymised donor data will be used in future publications arising from the study. Patients will be identified using only their unique registration number, patient initials on the case report form and correspondence between the trials office and the participating site. In addition, the patients are requested to give permission for the trials office to be sent a copy of their signed Informed consent form which will not be anonymised. This will be used to perform in-house monitoring of the consent process. Identifiable data will only be made available to authorised staff of the study sponsor, its authorised representatives and regulatory authorities. All patients will be consented specifically to enable data to be shared as detailed above. Confidentiality will otherwise be maintained throughout the trial and thereafter and data will be anonymised. On completion of the trial, data will be transferred to a secure archiving facility at the University of Birmingham, where data will be held for a minimum of 15 years and then destroyed.

**Table 3** Patient schedule of events

| Patient registration | Screening | Visit 1 Transplant Day 0 | Visit 2 Day 30 (+/−3 days) | Visit 3 Day 90 (+3 days) | Visit 4 Day 180 (+30 days) | Extended follow-up 12 months+24 months (+/− 30 days) |
|---|---|---|---|---|---|---|
| Informed consent | X | | | | | |
| Eligibility assessment | X | X | | | | |
| Patient history | X | X | | | | |
| Standard routine blood tests* | X | X | X | X | X | X |
| MELD (automatically calculated) | | X | | | | |
| UKELD (automatically calculated) | | X | | | | |
| Trial-specific additional patient samples blood and urine | | X | X | X | X | |
| PBMC collection | | X | X | X | X | |
| Liver biopsy 4 (see table 2) | | X | | | | |
| Quality of life questionnaire (EQ-5D-5L) | | X | X | | X | |
| Patient resource log at visit one discharge | | X | | | | |
| Adverse/Clinical events | X | X | X | X | X | X |
| Concomitant medications | X | X | X | X | X | X |
| MRCP | | | | | X | |

*Standard routine blood tests—FBC, urea, electrolytes, liver function tests, AST, GGT, eGFR, INR.
AST, aspartate transaminase; eGFR, estimated glomerular filtration rate; EQ-5D-5L, EuroQol 5 Dimensions 5 Levels; FBC, full blood count; GGT, gamma glutamyl transferase; INR, international normalised ratio; MELD, model for end-stage liver disease; MRCP, magnetic resonance cholangiopancreatography; PBMC, peripheral blood mononuclear cell; UKELD, UK end-stage liver disease.

## Electronic case report forms

Electronic case report forms (ECRFs) have been designed to capture as much, donor, perfusion and patient data as possible and feasible. The liver registration form and donor history form detail all that is relevant regarding the quality of the graft itself. The perfusion form enables collection of the perfusion parameters, biochemical data and the outcome of the perfusion. The patient registration and visit 1 forms will capture the demographics of the recipient as well as track the operative and postoperative course. Visits 2–4 are for patient follow-up.

## Trial organisational structure

The University of Birmingham will act as single sponsor this single-centre study. The trial is being conducted under the auspices of the CRCTU, The University of Birmingham according to their local procedures. The TMG will be responsible for the day-to-day running and management of the trial. Members of the TMG include the chief investigator, coinvestigators, project manager, trial management team leader, senior trial coordinator, trial coordinator, lead trial statistician and trial statistician. The TMG will have regular meetings during recruitment. The DMC will consist of independent clinicians Professor James Neuberger, Mr Gabi Oniscu and Professor Jacques Pirenne as well as an independent statistician, Mr Andrew Hall. Data analyses will be supplied in confidence to the independent DMC, which will be asked to give advice on whether the accumulated data from the trial, together with the results from other relevant research, justifies the continuing recruitment of further patients. The DMC will operate in accordance with a trial specific charter based on the template created by the Damocles Group. The DMC will meet at two scheduled time-points after the interim analyses as previously described (figure 2). An emergency meeting may also be convened if a safety issue is identified. The DMC will report directly to both the VITTAL Trial Management Group (chief investigator) who will convey the findings of the DMC to the trial steering group and funders/sponsor as appropriate or when specifically requested by these parties.

## Sources of funding

The VITTAL trial is funded by a grant awarded by the Wellcome Trust Health Innovation Challenge Fund (awarded December 2015).

## Trial status

Recruitment for the trial opened in October 2016 and recruitment is expected to last 24 months.

## DISCUSSION

The consequence of the escalating demand for liver transplantation is increasing waiting list mortality, and in many countries, patients are more likely to die while waiting for an organ than in the first year after their transplant.[45] The outcomes of high-risk livers are inferior to standard grafts and the difference is most noticeable within the initial 90 days. Indeed, severe early allograft dysfunction or primary non-function often trigger post-transplant sepsis and multiorgan failure and as consequence, livers with marginal features are often declined and discarded.

Our preliminary experience and pilot transplant series showed NMP-L can provide objective information regarding liver function and the VITTAL trial aims to produce robust data and validate our initial observations.

Several challenges were identified when designing the VITTAL trial with the foremost being to create a sound definition of a discarded liver. There is an undeniable variation in use of high-risk livers among the UK transplant centres which has been recognised and highlighted by NHSBT. The organisation published 'Taking organ transplantation to 2020', a strategy that aims to create greater consistency in the acceptance of organ offers and use of marginal livers across all centres.[46] To address this issue for this study purposes, every declined liver offered for enrolment into VITTAL has to meet also at least one of a list of predefined, constant inclusion measures, adopted in combination with a two-consultant system of macroscopic liver quality assessment.

The most important factor to consider while designing a trial that pushes the current boundaries of high-risk livers use is patient safety. Although we opted for liberal liver graft selection inclusion criteria, only low to moderate risk recipients are eligible to take part in this trial. Such an approach has been shown previously to be the safest and the most successful strategy for use of high-risk organs.[41 47] The intended recipients will be risk stratified and selected by the liver unit's liver transplant multidisciplinary team. Another important trial safety feature is its 3-stage adaptive design, introducing 2 interim safety analyses after completion of 3 and 11 transplants, respectively.

There are undoubtedly some livers that will not be salvageable or ever safe to transplant. It is important for the purposes of the trial to include organs that fail to meet the defined viability criteria to compare these with transplantable high-risk livers. The research work package linked with the trial was designed to identify sensitive point-of-care liver quality tests and propose novel biomarkers or panels associated with viable livers.

The primary end-point of 90-day patient survival has been chosen as it is a nationally accepted, monitored and continuously audited outcome following liver transplantation. Obviously, the graft survival rate is important and for the trial to truly be successful, patients who reach the primary end-point should have a VITTAL graft still in situ. This will be considered when the DMC monitor the results at the interim analyses.

As well as the study design, challenges with trial logistics were also identified. One of the previously unseen difficulties after discussion with the haematology team is the issuing of packed red cells matched to the intended recipient, potentially before the patient is admitted to hospital, to avoid delaying the start of the perfusion. When patients

are listed, they undergo a blood cross-matching process to identify blood group and the presence of antibodies. This sample is not held for longer than 7 days by the hospital and so if they are admitted for a transplant or require blood products for some other intervention, they have a new sample sent before those products are issued. In the case of the VITTAL trial, a perfusion may need to commence before the patient is admitted to hospital as they may have to travel some distance. Minimising the cold ischaemic time of marginal grafts is paramount to improve the chances of graft salvage. Therefore, in this scenario, blood is issued for the trial based on the results of the original sample and a repeat is sent when the patient is admitted to check they have not subsequently developed new antibodies. Blood product traceability is an important consideration and the blood products are documented to have been used in the device perfusate only and have not been used for recipient transfusion.

## ETHICS AND DISSEMINATION

The VITTAL Clinical trial is an academic investigator-led study involving a CE marked medical device. The device is being used outside it current CE mark and therefore has been reviewed by the MHRA UK and received a 'clinical investigation: no objection' (CI/2016/0031) letter: 3 August 2016. In addition, the study has undergone national ethical review in the UK and received national ethical approval from the London—Dulwich Research Ethics Committee (16/LO/1056) and the Health Research Authority. In addition to the above national regulatory approvals, the study has been reviewed by NHSBT service and received all appropriate local institution/NHS R&D approvals. The trial management team are also fully engaged in an academic collaboration with the device manufacture OrganOx as part of the management of this study.

The trial management team are fully committed to publishing (within 12 months of the end of the study) the results of this study in accordance with best clinical practice in an open-access, peer-reviewed medical journal irrespective of outcome. Any dissemination of results or publicity will be provided in a format which will not allow individual patients to be identified and confidentiality will be maintained throughout the process. The study management will be conducted in accordance with all applicable clinical trial regulations and managed centrally by the Drugs, Devices, Diagnostics and Biomarkers (D3B) trial management team— part of the Cancer Research UK(CRUK) clinical trials unit based in Birmingham in accordance with the quality management system. The results of the study will also be made available directly to study participants and specialist patient groups.

## SUMMARY

The presented VITTAL trial is the first clinical trial designed to objectively assess function of declined livers using NMP-L and subsequently transplanting viable grafts. It is hoped that the trial will identify a proportion of discarded organs that can be successfully transplanted and the generated data will provide objective and validated information that can be subsequently implemented in the process of acceptance and allocation of high-risk donor livers. This novel approach should improve consistency and increase use of marginal liver grafts without compromising recipient safety.

**Author affiliations**
[1]Department of Liver Unit, Queen Elizabeth Hospital, University Hospitals Birmingham NHS Foundation Trust, Birmingham, UK
[2]Department of Liver Biomedical Research Unit, National Institute for Health Research (NIHR), Institute of Immunology and Immunotherapy, University of Birmingham, Birmingham, UK
[3]Department of Cancer Research UK Clinical Trials Unit, Institute of Cancer and Genomic Sciences, University of Birmingham, Birmingham, UK

**Contributors** DFM (chief investigator) and HM (principal investigator) had the original concept of the VITTAL trial following on from a successful pilot study. DFM, HM, RWL, AK, CY and designed the VITTAL trial. RWL, MW, AK, and HM wrote the protocol and IRAS applications. DB, HM, SCA, PF and DFM reviewed all protocol versions. RWL, MW and DB (senior trials coordinator) submitted all REC, MHRA, HRA, NHSBT and local R&D applications. HM, AK, DFM and CY devised the statistical plan. TP, PM, JI, KR, HM and DFM are the transplant surgeons involved in the trial. RWL and MW wrote the patient information sheets, external trial information and patient CRFs. RWL, HM and DFM wrote the manuscript and all authors reviewed the final version.

**Funding** This trial is funded by a grant awarded by the Wellcome Trust Health Innovation Challenge Fund (awarded in December 2015). The grant application was a joint effort of the coapplicant team (Darius Mirza, Hynek Mergental, Simon Afford, David Adams, Peter Friend, Stefan Hübscher, Julian Bion, Christina Yap) and collaborators (Paolo Muiesan, Thamara Perera, Hentie Cilliers, Ricky Bhogal, Richard Laing, Amanda Kirkham, James Ferguson, Desley Neil, Amanda Smith, Steven Youngs, Darren Barton and Leslie Russell). The project was endorsed and supported by all the clinical staff at the Liver Unit of UHBFT and we would like to thank everybody for their hard work and dedication to the transplant programme. RWL, YB and SCA have received support from the NIHR Birmingham Liver Biomedical Research Unit.

**Disclaimer** The views expressed in this publication are those of the authors and not necessarily those of the NHS, the National Institute for Health Research or the Department of Health.

**Competing interests** RWL and YB receive salary as Wellcome Trust research fellows.

**Ethics approval** London-Dulwich (16/LO/1056)

**Provenance and peer review** Not commissioned; externally peer reviewed.

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
