## [Reviewer comments · BMJ Open]

ARTICLE DETAILS

TITLE (PROVISIONAL)	Viability testing and transplantation of marginal livers (VITTAL) using normothermic machine perfusion: A study protocol for an open label, non-randomised, prospective, single arm trial
AUTHORS	Laing, Richard; Mergental, Hynek; Yap, Christina; Kirkham, Amanda; Wilkhu, Manpreet; Barton, Darren; Curbishley, Stuart; Boteon, Yuri; Neil, Desley; Hubscher, Stefan; Perera, T; Muiesan, Paolo; Isaac, John; Roberts, Keith; Cilliers, Hentie; Afford, Simon; Mirza, Darius

VERSION 1 – REVIEW

REVIEWER	Prof. Irinel Popescu, MD, PhD, FACS, FEBS Fundeni Clinical Institute, Center of General Surgery and Liver Transplantation Bucharest, Romania
REVIEW RETURNED	29-May-2017

GENERAL COMMENTS	The presented VITTAL trial, analyzing the value of viability testing of marginal livers using normothermic ex-situ perfusion, is the first clinical trial designed to objectively assess the function of declined livers and subsequently transplanting viable grafts. The trial is aimed to identify a proportion of discarded organs that can be successfully transplanted. This novel approach should increase the use of marginal liver grafts without compromising recipient safety. This is a well-designed study, and a properly structured paper, written in good English, and should be considered for publishing in the present form.
---

REVIEWER	Prof. Dr. med. M.K. Angele, M.D., F.A.C.S. Lead Consultant Management of surgical oncology Department of General, Visceral, Vascular and Transplant Surgery Ludwig-Maximilians-University, Munich Campus Grosshadern Marchioninistraße 15 D-81377 Munich Germany
REVIEW RETURNED	31-May-2017

GENERAL COMMENTS	The proposed study aims to investigate the role of ex-situ normothermic machine perfusion of the liver provides helpful
---

	information on the use of currently deemed untransplantable livers. The study addresses an interesting subject. Nonetheless, I have a few concerns that should be addressed by the authors. The inclusion criteria “untransplantable liver” is a very heterogeneous parameter. This criterion should be more defined to make the results more reliable. Organ survival will be compared with a general patient collective undergoing liver transplantation. Thus, it remains unclear what MELD scores and diagnosis are accepted for transplantation. This should be specified in the study proposal and considered as potential confounders for analysis. In summary, the study proposal addresses an interesting subject and therefore should be considered for publication.
--	---

VERSION 1 – AUTHOR RESPONSE

Reviewer 1

Dear Professor Popescu,

We thank you for your positive remarks and echo your sentiments that this study has the potential to increase the number of transplantable donor livers.

Reviewer 2

Dear Professor Angele,

Thank you very much for your constructive comments and we have taken the opportunity to clarify several points.

“The inclusion criteria “untransplantable liver” is a very heterogeneous parameter. This criterion should be more defined to make the results more reliable.”

The term “untransplantable” appears in the abstract and we have changed this to reflect that we have defined objective criteria that denote the “extended criteria” or “marginal” nature of these organs. In addition, the livers that are being included must have been decline for use by all UK transplant centres and we have added this to the abstract and go into more detail in the manuscript. We agree that the trial must clearly show that we are able to transplant livers that would not have been transplanted because they are currently perceived to be too high risk to use.

“Organ survival will be compared with a general patient collective undergoing liver transplantation. Thus, it remains unclear what MELD scores and diagnosis are accepted for transplantation. This should be specified in the study proposal and considered as potential confounders for analysis.”

The patients from the study will be matched using the demographics and characteristics detailed on page 15 of the manuscript; including but not exclusively the patient characteristics age, sex, BMI, MELD, UKELD, aetiology and donor liver characteristics including graft type and sex of donor.

We have clarified in the manuscript that we have no upper or lower limits for MELD score as this system is not used to guide organ allocation in the UK.

We do have an exclusion criteria upper limit of 62 for UKELD in keeping with the NHSBT LAG criteria for graft sharing in high risks recipients in the North East of the UK. We have also clarified that we will not exclude any specific chronic liver disease aetiology however fulminant hepatic failure and retransplant recipients are excluded. We will investigate and adjust for potential confounders in our analysis (detailed on pages 21 and 22 of the manuscript).

VERSION 2 – REVIEW

REVIEWER	Prof. Irinel Popescu, MD, PhD, FACS, FEBS Fundeni Clinical Institute, Bucharest, Romania
REVIEW RETURNED	04-Jul-2017

GENERAL COMMENTS	The reviewed paper responded to all comments of the reviewers, including the corresponding modifications in the text. It's a well-designed study and properly structured paper, written in good English. I therefore recommend to be considered for publishing in the present form.
---